# PACSAB Server: A Web-Based Tool for the Study of Aggregation and the Conformational Ensemble of Disordered and Folded Proteins

**DOI:** 10.3390/ijms25116021

**Published:** 2024-05-30

**Authors:** Agustí Emperador

**Affiliations:** Department of Physics, Universitat Politècnica de Catalunya, B4-B5 Campus Nord, Jordi Girona 1-3, 08034 Barcelona, Spain; agusti.emperador@upc.edu

**Keywords:** protein association, protein aggregation, intrinsically disordered proteins, structural ensemble

## Abstract

We present in this article the PACSAB server, which is designed to provide information about the structural ensemble and interactions of both stable and disordered proteins to researchers in the field of molecular biology. The use of this tool does not require any computational skills as the user just needs to upload the structure of the protein to be studied; the server runs a simulation with the PACSAB model, a highly accurate coarse-grained model that is much more efficient than standard molecular dynamics for the exploration of the conformational space of multiprotein systems. The trajectories generated by the simulations based on this model reveal the propensity of the protein under study for aggregation, identify the residues playing a central role in the aggregation process, and reproduce the whole conformational space of disordered proteins. All of this information is shown and can be downloaded from the web page.

## 1. Introduction

Despite the continuous increase in computational power, biomolecular processes extending beyond the millisecond time scale, like the initial stages of protein aggregation or the conformational sampling of large disordered proteins, remain beyond the capabilities of standard molecular dynamics (MD) simulations with explicit representation of solvent.In the case of stable folded proteins, where a native structure of the protein is well defined, elastic network models and structure-based potentials have produced very good predictions for their flexibility pattern with an extremely low cost in computational power compared with standard MD. These models are at basis of web tools like ElNemo [1], FlexServ [2] or WEBnm@ [3], which allow for the production of the flexibility pattern of a protein, and iMODS [4], which allows for the description of collective motions of macromolecules and generate realistic conformational transition pathways. Unfortunately, these models cannot be applied to disordered proteins or multiprotein systems as in these cases, a native configuration of the system is not defined.

Standard molecular dynamics simulations are commonly created with software packages, many of them being free like GROMACS [5] (https://www.gromacs.org/ (accessed on 23 May 2024))or NAMD [6] (http://www.ks.uiuc.edu/Research/namd/ (accessed on 23 May 2024)), that users have to install onto their own computational resources. So far, a limited number of web tools, like MDWeb [7] (https://mmb.irbbarcelona.org/MDWeb/ (accessed on 23 May 2024)) or WebGRO (https://simlab.uams.edu/ (accessed on 23 May 2024)) offer to researchers without any computational skills the possibility to run molecular dynamics simulations in remote servers. The MERMAID web tool [8] (https://molsim.sci.univr.it/mermaid/begin.php (accessed on 23 May 2024)) is based on the coarse-grained MARTINI model [9], which makes it much more suitable for simulating large macromolecular systems; however, like standard MD methods, it uses explicit representation of the solvent, which is less efficient than the implicit solvent approach for the confomational sampling of multiprotein systems. Due to these limitations, web tools designed to address the protein aggregation problem like AGGRESCAN [10] (http://bioinf.uab.es/aggrescan/ (accessed on 23 May 2024)), Waltz [11] (https://waltz.switchlab.org/ (accessed on 23 May 2024)), Tango [12] (http://tango.crg.es/ (accessed on 23 May 2024)), AGGRESCAN3D [13] (https://biocomp.chem.uw.edu.pl/A3D2/ (accessed on 23 May 2024)), Solubis [14] (https://solubis.switchlab.org/ (accessed on 23 May 2024)), or Camsol [15] (https://www-cohsoftware.ch.cam.ac.uk (accessed on 23 May 2024)) use bioinformatics-based methods instead of MD-based methods (for a collection of web tools for the study of protein aggregation, see the review article [16]).

Implicit solvent models can achieve a fast sampling of the conformational space of the protein system in MD simulations thanks to the lack of friction with solvent molecules, which produces fast diffusion and a much faster sampling than in explicit solvent simulations, with a large speedup factor for implicit solvent simulations of systems without kinetic barriers [17]. Recently, we developed a detailed coarse-grained model [18] capable of accurately predicting the conformational space of small disordered proteins and the protein–protein recognition of both stable and disordered proteins thanks to the use of the implicit solvation approach: the PACSAB (*Pairwise Additive Coarse-grained Sidechain and Atomistic Backbone*) protein model uses an atomistic description of the protein backbone in order to account for backbone hydrogen bonding, which plays a fundamental role in the structural ensemble of disordered proteins, as well as a coarse-grained representation of the amino acid side chains. We present in this paper the PACSAB web tool (https://pacsab.upc.edu), whose purpose is to produce molecular dynamics trajectories based on the PACSAB coarse-grained model, and we analyze these trajectories to provide information about the conformational space of the system and protein–protein interactions.

To simulate a protein solution, in our PACSAB web tool, two copies of the protein under study are placed inside a simulation box with periodic boundary conditions. The calculations are computed in a back-end compute server, and the results of the analysis are uploaded to the web server, which delivers to the user information about the association dynamics of the protein system, contact maps showing which residues of the protein sequence play an important role in aggregation, and the radius of gyration distribution and RMSD (root mean squared deviation) along the trajectory, characterizing the conformational ensemble of the protein. The PACSAB server offers the possibility to change the parameterization of the force field used in the simulations in order to obtain a conformational ensemble in better agreement with the available experimental information or to obtain higher statistics of association and dissociation events.

## 2. Results

We present here the cases shown in the examples section of the web, a selection of the simulations generated so far with the PACSAB server: the small protein villin, whose association dynamics were thoroughly studied elsewhere with atomistic molecular dynamics simulations, the stable protein ubiquitin, with a clear dimerization interface revealed from experimental studies, and two intrinsically disordered proteins (IDP): the non-aggregating ACTR and the aggregating amyloid-β peptide Aβ40. In all cases, we have used the same parameters for the force field that are set as default values when entering the parameters for a simulation in the PACSAB web page.

We consider that one of the main applications of the PACSAB server is to identify the residues playing a key role in the protein association process. In our previous work [19], our aim was to exactly reproduce the fraction of monomers in a solution of ubiquitin at a given concentration, and we used that system to calibrate the PACSAB force field parameters. Here, on the contrary, we want to obtain high statistics of the transient dimeric structures; that is why we used a slightly less hydrophobic parametrization of the force field in our server. A 3% decrease in the Van der Waals term was enough to increase the dissociation rate of the transient dimers found in the simulation of the ubiquitin solution, producing bigger statistics of association events and a consequently wider collection of transient dimers.

### 2.1. Villin

We produced a simulation of an 8.5 mM solution of the 36-residue-long villin HP36 (PDB IB 1VII), a small α-helix protein used for the benchmarking of atomistic force fields. Despite villin being known to weakly associate at high concentrations (32 mM), atomistic explicit solvent molecular dynamics simulations with the standard TIP3P water model [20] produce a straightforward aggregation of the proteins [21]. Best et al. [22] simulated an 8.5 mM solution of villin using a more accurate water model, TIP4P/2005 [23], to correct the spurious aggregation obtained with the TIP3P water model. We show in Figure 1 the minimum distance between the two villin molecules in our simulation, wherein we use structure-based potentials to stabilize the tertiary structure of this highly flexible protein. This figure is to be compared with Figure 4 of the work by Best et al. [22]. We find multiple associations followed by fast dissociations, which keeps almost all of the villin molecules in the monomeric state during the entire simulation.

### 2.2. Ubiquitin

Next, we show the results of the simulation of the 76-residue-long ubiquitin (PDB ID 1UBQ). NMR experiments [24] showed that ubiquitin forms transient, low-affinity, noncovalent dimers, defined by a large interface where many relative orientations are possible. These measurements allowed for the identification of the binding interface of ubiquitin, a β-sheet surface formed by the residues 4–12, 42–51, and 62–71. The same experimental study showed that at 5 mM concentration, one half of the ubiquitin molecules remain in the monomeric state, while the rest form dimers or higher-order oligomers.

Abriata et al. [25] simulated a 5 mM solution of ubiquitin with explicit solvent molecular dynamics, finding a collapse of the molecules into an aggregate. As explained by the authors, this spurious result is due to the inaccuracy of the TIP3P water model that they used in the MD simulations. Motivated by their assessments, we studied in a previous work [26] the dimerization of ubiquitin at this concentration using the accurate TIP4P/2005 water model in explicit solvent molecular dynamics simulations.

Now, we show in Figure 2 the results of our simulation of a system with 5 mM concentration with PACSAB. In order to conserve the tertiary structure of the ubiquitin molecules, we chose to add structure-based potentials to the intramolecular interactions. The figure shows that the system undergoes several association events that form transient oligomers of different lifetimes and eventually forms a dimer that persists during 300 ns. Comparison with Figure 1 of our previous work [26] makes evident that in our simulation the ubiquitin molecules move much faster than in explicit-solvent MD. This diffusion, which is faster than the real diffusion of the ubiquitin molecules, allows the PACSAB model to sample many association events in short trajectories, making it a very good method for the conformational sampling of a multiprotein system.

We also plotted in Figure 2 the intermolecular contact maps showing the residue–residue contacts, accounting up to 120 ns and for the entire simulation (note that the color scale is different in the two graphs). The X- and Y-axes are the index of the residues along the protein sequence in each molecule. The 120 ns contact map includes only the contacts of the dimer formed at around 110 ns, which involves residue 57 and is around the protein sequence, therefore being outside the binding interface. On the contrary, it can be observed that the main contribution to the contact map of the whole simulation is from the residues around 64 and 45, belonging to the binding interface. These contacts appear in the dimer formed at 400 ns, which remains unbroken for 300 ns.

### 2.3. ACTR

Another example shown in the web page is the simulation of a solution at a high concentration (12 mM) of the 46-residue-long intrinsically disordered protein ACTR (activator for hormone and retinoid receptors; its structure when bound to its binding partner is PDB ID 1KBH), which has a very low propensity for aggregation due to its highly hydrophilic sequence. Best et al. [22] studied the conformational ensemble of this disordered protein, finding that the best results were obtained with the TIP4P/2005 water model, although the ensemble was still too collapsed when compared with the radius of gyration of 2.5 nm measured experimentally by SAXS [27].

In a previous work [26], we found that explicit solvent molecular dynamics simulations of this protein at 10 mM concentration with the standard TIP3P water model produced straightforward aggregation, but when the more accurate TIP4P/2005 water model was used, the molecules underwent frequent dissociation events, staying monomeric most of time.

In order to simulate this IDP, we had to generate a random coil structure starting from the RCSB Protein Data Bank structure of ACTR when it was bound to its binding partner, the NCBD protein, where ACTR shows three helices in sections 5–17, 24–31, and 34–40 of its sequence. To accomplish this, we ran a 10 ns simulation with an extremely hydrophilic force field parametrization, i.e., a very low Van der Waals factor and very high solvation factor (see the Methods section for a description of the force field), and weakening the hydrogen-binding term.Afterwards, we restarted the simulation with the default PACSAB force field parameters.

Figure 3 shows that, despite many collisions occurring during the simulation, no dimers are formed. In the intramolecular contact map, a subtle local refolding is observed on the ACTR chain, reminiscent of the first and third helices of ACTR when it is bound to its binding partner. In our study of ACTR:NCBD binding in a previous work [19], we found that the association is made possible by the refolding of ACTR to its helical structure, followed by the recognition of the binding interface. The process of the refolding of a disordered protein into its bound stable structure upon binding follows a mechanism that is generally a mixture of the two ideal cases of induced fit and conformational selection [28].

The size of the conformational ensemble of ACTR had been estimated from SAXS measurements to have a radius of gyration $R_{g}\approx 25$ Å. The ensemble obtained by Best et al. [22] in their explicit solvent molecular dynamics simulations with TIP4P/2005 water for ACTR showed a distribution of the radius of gyration of the protein, whose maximum is around 14 Å. We show in Figure 4 the distribution of $R_{g}$ found with our simulation based on PACSAB, which shows its maximum around 19 Å, closer to the experimental observations. This distribution coincides with the distribution we obtained in our previous work, showing that the slight change that we introduced in the PACSAB force field parameterization had no effect on the structural ensemble of this disordered protein.

### 2.4. Aβ40

Finally, we show the results for the 40-residue-long peptide Aβ40 (PDB ID 2LFM). Amyloid-β peptides, whose different alloforms are produced from cleavage of the amyloid precursor protein, form amyloid plaques in the brain of people with Alzheimer’s disease, and the oligomers of amyloid-β are considered to be the main neurotoxic agent in Alzheimer’s disease [29,30]. The Aβ40 peptide is disordered in aqueous environment but shows an α-helical structure when embedded in a lipid-like hydrophobic environment, and this is the structure that can be downloaded from the RCSB Protein Data Bank. So, in this case, we used the same procedure as for ACTR: a short simulation with an altered, extremely hydrophilic force field parameterization to obtain the random coil structure, then restart the simulation with the PACSAB force field parameters.

Unlike ACTR, Aβ40 has a very hydrophobic sequence and therefore a high propensity for aggregation. For this reason, we simulated a solution with a much lower concentration, 0.5 mM, which involved a simulation box of 188 Å. Taking into account that the encounter frequency between two peptides in water is on the order of 10^10^ s^−1^ M^−1^ [31], the average collision time at a 0.5 mM concentration would be on the order of 200 ns, which is impossible to afford with an explicit solvent simulation that would involve millions of water molecules in such a large simulation box. This is one case in which the only way possible is to use the implicit solvation approach, which produced very revealing results in the study of the aggregation process in an Aβ40 solution at a 0.77 mM concentration [32].

We show in Figure 5 the distance between the two Aβ40 peptides during our simulationof the 0.5 mM solution. Apart from some collisions between the two molecules, the first association happens at around 800 ns, and afterwards, the two peptides do not dissociate, which is consistent with the extremely aggregating character of this peptide. We also show in the figure the intramolecular contacts, where the incipient formation of an α-helix region can be observed around the residues 12 and 17. The Aβ40 molecule undergoes this transition from the random coil state to an amorphous collapsed state due to the high hydrophobicity of its sequence. It is important to take into account that the intermolecular contacts that we find (visible in the web page, examples section) is just the interface of the complex formed by the two Aβ40 at t ≈ 800 ns, which does not change for the rest of the simulation. To obtain an intermolecular contact map reflecting the propensity of each residue to form intermolecular contacts, therefore playing an important role in the aggregation process, it would be necessary to have high statistics of the dimer structures, and this is not the case here, as we found just one dimer.

## 3. Discussion

The web tool presented in this paper is the first one based on molecular dynamics simulations of a coarse-grained protein model in an implicit solvent environment.This makes it suitable for studying problems like protein association or the structural ensemble of disordered proteins, which cannot be addressed with other existing web tools based on molecular dynamics simulations due to the lower sampling efficiency when an explicit representation of solvent is used, even with a coarse-grained model of the system. An important application of our tool is to identify the residues playing a key role in protein association. In order to gain efficiency in the sampling of the protein–protein interfaces in transient dimers, we slightly changed the parameterization of the PACSAB force field of our previous work [26]. This change, which strongly increased the dissociation rate and statistics of transient dimers, did not affect the results for the structural ensemble for the disordered protein ACTR, as can be checked by comparing the distribution of the radius of gyration obtained here with that obtained in our previous work [26].

The simulation of the highly aggregating Aβ40 peptide did not produce any statistics for transient dimers as no dissociation was found, in agreement with the real behavior of this peptide, whose sequence is highly hydrophobic. In order to obtain a contact map that identifies the residues playing a key role in the aggregation of Aβ40, a big statistics of the dimeric structures is needed. To achieve this, it would be necessary to run enhanced sampling simulations like replica exchange molecular dynamics (REMD), something which proceeds far beyond the scope of this web tool. Apart from this, our simulations only show the very beginning of the aggregation process because, as explained in the review article [31], the oligomers have to overcome a free-energy barrier before reaching the stable amyloid β-strand structure of the amyloid fibrils observed in experiments. In studies of very small fragments of the amyloid β peptide, it was observed that the peptides first underwent a hydrophobic collapse (due to the absence of dissociation events), and in a longer time scale, the oligomer evolved into an ordered β-sheet structure. This time scale is orders of magnitude longer than the time scale spanned by our simulations.

## 4. Materials and Methods

The PACSAB web tool is composed of two parts: a web server and a compute server. The web server is written in PHP and implemented as a web-based interface, acting as a front end where the user can submit MD simulations and obtain the results of the analysis of the trajectory as the MD simulation runs. It is important to emphasize that, unlike the web tools based on bioinformatics, our method is based exclusively on the PACSAB force field, so we do not use structural libraries or datasets. After uploading the coordinates of the protein, the web server sends these coordinates to the compute server, then several bash scripts call our own FORTRAN code that generates the PACSAB model of the protein, create a copy of the protein, and finally run the molecular dynamics simulation with our PACSAB code, implemented in FORTRAN. As explained in our previous works [18,33], to model a protein solution, we use a system formed by 2 molecules of the protein in a simulation box with periodic boundary conditions whose side length *L* corresponds to the desired concentration *C*, which in this case is $C=2/L^{3}$. The concentration in M (mol/dm³) would be therefore $C=2/(N_{A}L^{3})$ M, where $N_{A}=6.02\times 10^{23}$, and *L* is expressed in dm.

The PACSAB tool allows for the simulation of proteins with a maximum of 200 residues, as a higher number would involve a simulation that is too slow. In order to limit the size of the trajectory files, which are conserved in the hard disk of the compute server, it writes 1000 frames for the full trajectory, regardless of the trajectory length. The user defines the frame-to-frame time $t_{F}$ (≥10 ps), so in the case of very dilute systems where long simulations must be run to find protein–protein association events, the user can define a large $t_{F}$. The running time of the simulation will depend on the trajectory length, as occurs with a standard MD simulation. The PACSAB server allows for an on-the-fly analysis of the simulations, i.e., it is not necessary to wait until the end of a simulation to see the results; the user can trigger the analysis process or even stop the simulation at any moment. The trajectories, which are kept in the hard disk of the compute server, can also be restarted afterwards.

In order to obtain access to the PACSAB web tool, a user account must be requested by sending an email to the corresponding author of this paper. Once the account is activated, the user can enter their own workspace and choose to:Submit a new simulation (if the user has no other simulation running);Restart a previous simulation (if the user has no other simulation running);Observe the results of a simulation (either running or finished);Run an analysis of a simulation (either running or finished).

The structure of the web tool is illustrated in the flowchart of Figure 6. The red rectangles in the flowchart mark actions performed by the compute server: prepare the PACSAB model structure of the system, analyze a simulation, or run a simulation. In the case of no simulation running, the user’s home page also shows a section where the user can submit a new simulation through a web form; the user has to upload the protein structure file with the coordinates of the protein in the GROMACS [5] format (.gro file) (see information in https://www.gromacs.org for more details), which is the format that the simulation machinery of the compute server is able to understand (instructions on how to generate the .gro file from an RCSB Protein Data Bank structure (https://www.rcsb.org/ (accessed on 23 May 2024)) are given in the web page). The user is also asked for the frame-to-frame time $t_{F}$, the size of the simulation box (determined by the concentration of the solution to simulate), and the offset of the copy of the protein (as the system simulated consists of two copies of the protein in a box with periodic boundary conditions). The user can choose to use structure-based potentials, which is useful in the simulation of stable proteins (this would not make sense in the case of disordered proteins as a native structure of the protein is not defined in this case), or an unfolding force field parametrization with very hydrophilic values of the force field and very low hydrogen bonding energy, very useful for generating random coil structures to be used as starting conformations for the simulation of disordered proteins, whose structure in aqueous solution is not well defined. The user can also modify the parameters of the PACSAB force field, whose values are set as default in the form.

The details on the implementation of the PACSAB model are explained in our previous works [18,33]. The side-chain bead interaction potentials in the PACSAB force field are optimized to produce the correct association/dissociation equilibrium of peptides and are evaluated as the addition of the interaction potentials between the atoms included in the coarse-grained beads. The PACSAB force field fundamentally depends on two parameters, which are the strengths of the van der Waals and the solvation terms [18]:$$V\left(r_{ij}\right)=\omega_{vdW}V_{vdW}\left(r_{ij}\right)+\omega_{solv}V_{solv}\left(r_{ij}\right)$$

The hydrophobicity of the force field increases with the Van der Waals factor $\omega_{vdW}$ and decreases with the solvation factor $\omega_{solv}$. In order to take into account the tendency of disordered proteins to form local secondary structure elements (mainly helices formed by residues close in sequence), the values of $\omega_{vdW}$ and $\omega_{solv}$ are interpolated between a “short-range” value for nearby residues, which are more hydrophobic, and a “long-range” value for distant residues, and its value depends on the distance of the residues, as detailed in the methods section of our previous work [33]. The same can be performed for the hydrogen bonding strength, although in the PACSAB force field, we set the same value of the hydrogen energy for residues close and distant in sequence.

Once the protein coordinate file has been uploaded, the server creates a copy of the molecule, and the user is asked to download the PDB coordinates file of the whole simulation box to confirm that the relative position of the two molecules is correct, as shown in Figure 7. The user then decides if the starting configuration is good for beginning the simulation or if the setup has to be discarded.

In the case of a restarted simulation, the starting configuration of the system is just the last frame of the previous simulation, but the user can change the force field parameters. This is especially useful for the simulation of disordered proteins: Because the available structures in the RCSB Protein Data Bank are those of the protein folded in a hydrophobic environment or when it is bound to its binding partner, an initial simulation with an unfolding force field parameterization can be performed. A short simulation of 10 ns ($t_{F}$ = 10 ps) is enough to move the protein conformation to the random coil structure of a disordered protein in solution. Then, the simulation can be restarted selecting the standard PACSAB force field parameters (note that the system takes the parameters of the previous simulation by default) and increasing $t_{F}$ to produce a simulation as long as needed to sample the conformational space of the disordered protein.

When the user chooses to run the analysis of a simulation, the compute server executes a series of scripts to calculate the RMSD and radius of gyration of the molecules with the cpptrajprogram of the AmberTools package (https://ambermd.org/AmberTools.php (accessed on 23 May 2024)), and it generates the figures shown in the results page with gnuplot (https://gnuplot.sourceforge.net/ (accessed on 23 May 2024)). The minimum intermolecular distance along the simulation and the number of intramolecular and intermolecular residue–residue contacts are calculated with our own FORTRAN code.

When the user chooses to view the results of the simulation, the figures created from the analysis are uploaded from the compute server to the web server. In the results page, it shows:The minimum molecule–molecule distance along the trajectory;The radius of gyration distribution, which describes the conformational ensemble of a disordered protein and can be confronted by experimental measurements;Contact maps: intramolecular contacts and intermolecular contacts, very useful for identifying residues that play an important role in the association of aggregating proteins. In the case of IDPs, the formation of the local structure in the coil (partial refolding) can be observed in the intramolecular contact map;Evolution of the RMSD along the trajectory for each of the two proteins. This is useful in the case of stable proteins if no structure-based potentials are used to study how the interaction of the proteins can induce conformational transitions;Evolution of the radius of gyration with time. It is interesting in the case of disordered proteins to observe the collapse of the protein when starting the simulation from a random coil generated with altered force field parameters;2D-RMSD: It interesting to observe the formation of structural clusters during the simulation.

To allow the user to further analyze the trajectory with tools external to the PACSAB web, the results page allows the user to download the data of the minimum distance along the trajectory, the RMSD and radius of gyration for each of the two molecules, and the last frame of the simulation. There is also a link to download the complete trajectory with a backbone-level resolution (excluding the coarse-grained side chains of the PACSAB model). The atomistic resolution of the backbone allows us to study the evolution of the secondary structure along the trajectory, which is interesting in the case of disordered proteins. Because the trajectory is a heavy file, the transfer of this file from the compute server to the web server is not instantaneous, and the user has to wait some time until the download link appears.

In the home page of the server, there are links to the results of some example cases, which can be viewed without a user account.

## 5. Conclusions

The results obtained for the test cases shown in this work demonstrate that the PACSAB web tool presented here is able to produce a realistic description of the association of folded and disordered proteins. The model used by the PACSAB server is able to identify the binding interface of ubiquitin, a molecule whose interface in dimerization is well known from experimental studies; in our simulation, dimers whose interfaces coincide with the real interface are much more stable than dimers whose interfaces do not completely coincide with the real one.

The PACSAB model also correctly evaluates the aggregation propensity of different disordered proteins, and we have checked that for the disordered protein ACTR it produces a structural ensemble that is in much better agreement with experimental observations than the results obtained from standard explicit solvent molecular dynamics simulations.

The accuracy of the results produced by the PACSAB model, along with the ease of use of the PACSAB web interface and its fastness in describing the structural ensemble and aggregation dynamics of proteins, makes us believe that this can become a useful tool for researchers interested in protein aggregation and the function and interactions of small disordered proteins. Our perspective for the near future is to broaden the applications of the web tool, allowing for the simulation of systems of two different proteins, which will allow for the study of association between different proteins, both stable and disordered.

## Figures and Tables

**Figure 1 ijms-25-06021-f001:**
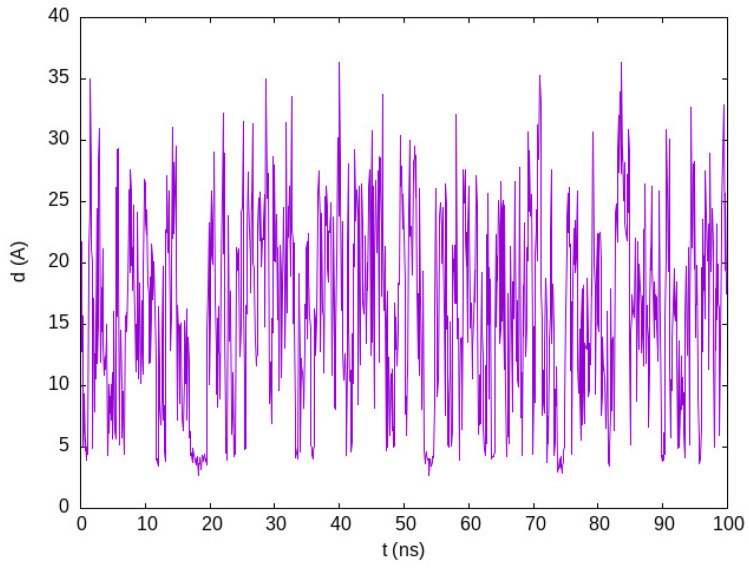
Minimum distance between two villin molecules in the simulation of a villin solution at 8.5 mM concentration, as shown in PACSAB web.

**Figure 2 ijms-25-06021-f002:**
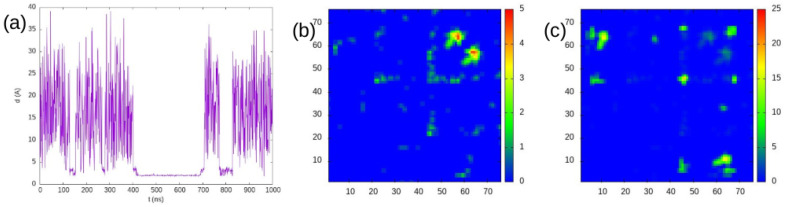
(**a**) Minimum distance between two ubiquitin molecules in the simulation of a ubiquitin solution at 5 mM concentration. (**b**) Intermolecular contacts accounted for until 120 ns of the simulation. (**c**) Intermolecular contacts accounted for during the entire simulation (note the difference between the color scales of (**b**,**c**)).

**Figure 3 ijms-25-06021-f003:**
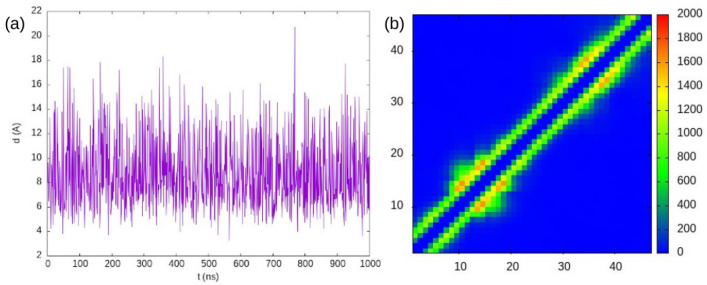
(**a**) Minimum distance between two ACTR molecules in the simulation of an ACTR solution at 12 mM concentration. (**b**) Intramolecular contacts.

**Figure 4 ijms-25-06021-f004:**
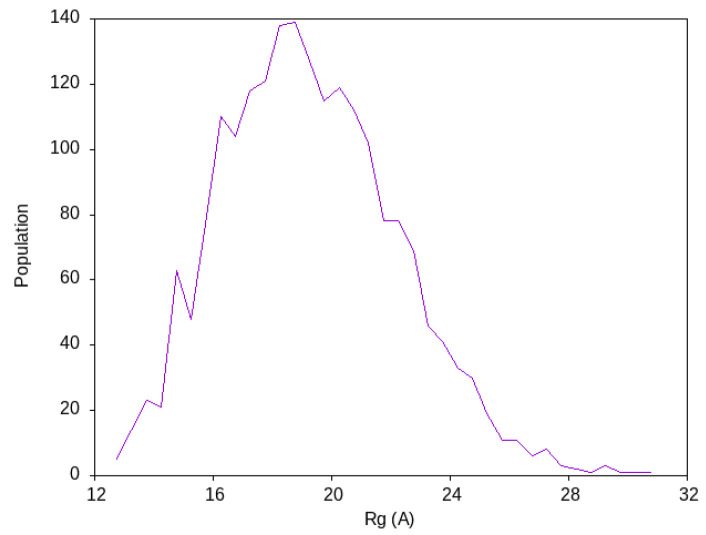
Radius of gyration distribution of ACTR in solution at a 12 mM concentration, as shown in PACSAB web.

**Figure 5 ijms-25-06021-f005:**
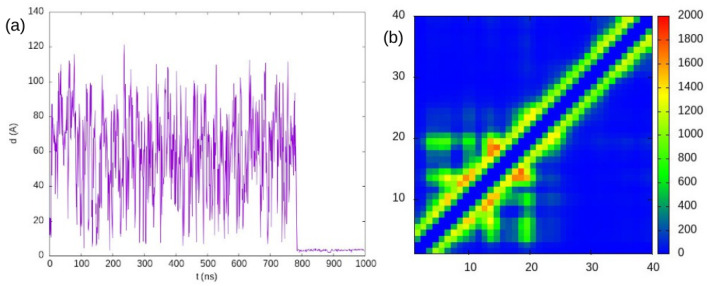
(**a**) Minimum distance between two Aβ40 molecules in the simulation of a solution at 0.5 mM concentration. (**b**) Intramolecular contacts.

**Figure 6 ijms-25-06021-f006:**
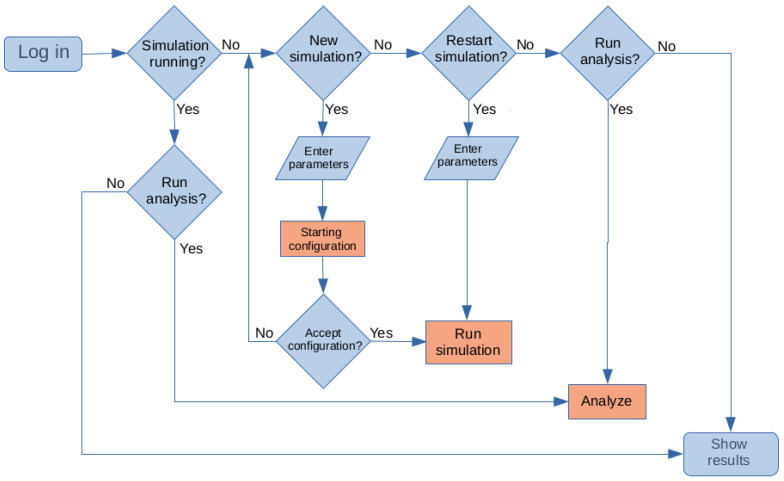
Flowchart of the PACSAB web tool.

**Figure 7 ijms-25-06021-f007:**
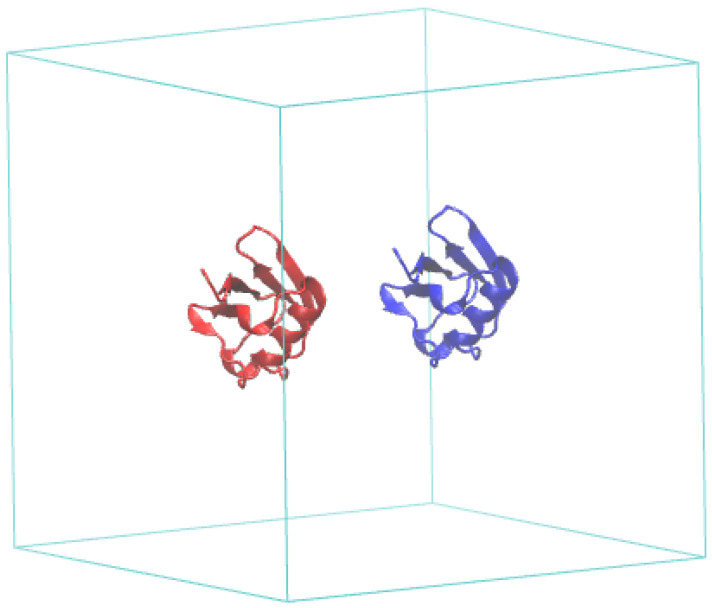
Starting system configuration generated by the PACSAB server for the simulation of a ubiquitin solution with a concentration of 5 mM, which corresponds to a box side of 87 Å.

## Data Availability

Data is contained within the article.

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
