# Peer review of "PACSAB Server: A Web-Based Tool for the Study of Aggregation and the Conformational Ensemble of Disordered and Folded Proteins"

_ijms, 2024, doi:10.3390/ijms25116021_

Round 1

Reviewer 1 Report

Comments and Suggestions for Authors

This manuscript presents a web server implementing the PACSAB simulation method developed by the author, and can be used to study the self-interaction of proteins and disordered peptides. The manuscript is short as the PACSAB forcefield and algorithm was described in earlier publications. The web server seems to be a useful tool, although its application is limited by the fact that only homodimerization can be studied, and the chain length is limited to 200 residues. Perhaps a downloadable version without these limitations could also be offered.

I think that a few improvements to the manuscript are needed before publication:

1. Please add the PDB entry codes of the starting structures you used for the example proteins (villin, ubiquitin, ACTR, Aβ40).

2. The most relevant structures generated by the simulations should be shown.

3. Lines 152-154: "For this reason we have slightly modified the force field parameters respect to the PACSAB force field we used in our previous work [9] to increase the dissociation rate" - Please explicitly describe what this slight modification was.

4. Ref. 9 (earlier work by the author) presents PACSAB simulations for ubiquitin, ACTR, and Aβ40. I think that a brief comparison between those earlier simulations and those presented in this manuscript is warranted.

5. For ubiquitin, was a structure-based potential used? In Ref. 9, this was necessary to prevent a distortion of the structure, but the current manuscript does not indicate whether it was used.

6. For ubiquitin, villin, and ACTR, the manuscript briefly discusses earlier simulation studies, but there is hardly any discussion of previous work for Aβ40, although it is a highly studied protein. Please include more comparison with previous work. Also, a dimeric structure with an α-helical region was found for Aβ40. How does this compare with earlier simulations? Can we expect the method to reproduce β-amyloid fibril formation (aggregation by β-strands)? Perhaps Aβ42 could be tried, which has a higher propensity to form fibrils.

7. Line 60: the minimum frame-to-frame time is 10 ps according to the manuscript. Is there a maximum frame-to-frame time? How does the running time depend on the frame-to-frame time? What were the frame-to-frame times in the simulations presented as examples in the manuscript, and what were the total running times (in real time on the compute server)?

I recommend the manuscript for publication after these improvements.

Comments on the Quality of English Language

Minor corrections are necessary, e.g. this sentence (line 152-153):

"For this reason we have slightly modified the force field parameters WITH respect to the PACSAB force field" 

Author Response

I thank the referee for his/her careful reading of the manuscript. His/her comments have increased the quality of the manuscript.

I send the referee the credentials to log in the web tool, just in case he/she did not get it from the editors in the initial submission of the manuscript:

User: 'ref1'

Password: 'SCFtcm137'

I have introduced the changes and addressed the issues commented by the referee:

1- I have added in the manuscript the PDB ID of the structures used for the simulations.

2- Regarding the most relevant structures, these are shown in the web page. These are the native conformations of the stable proteins (villin and ubiquitin) and regarding the disordered proteins, there is no representative structure, since ACTR in solution is a random coil and AB40 is highly amorphous. I have shown in the web page the structure of ACTR when complexed to its binding partner and that of AB40 within a hydrophobic environment, where an helical structure is stabilized

3- I have improved the explanation of the initial version of the manuscript, which was a little confusing. Now this is explained in the second paragraph of the Results section (lines 182-190)

4- I have introduced a comparison between the results obtained here and in the earlier work in the Results section, lines 182-190 and 262-264

5- We used a structured-based based potential in the simulation of ubiquitin and and that of vilin. I have written this in the new version of the manuscript.

6- A thorough comment addressing the issues raised by the referee has been added to the manuscript, it is in the second paragraph of the Discussion section (lines 308-319). Regarding the possible simulation of AB42, no differences would have been found respect to AB40 in the time scale that the simulation spans, since the differences emerge at longer time scales, when fibrils start forming after the lag phase.

7- I have rewritten this part of the manuscript, which was indeed a little confusing (lines 80-85). The simulation time depend on the trajectory length, and the frame-to-frame time is just the trajectory length/1000 (the system writes 1000 frames)

Reviewer 2 Report

Comments and Suggestions for Authors

In this manuscript, the authors present PACSAB server, providing information about the structural ensemble and the interactions of both stable and disordered proteins for molecular biology reserach. The advantage of this tool is that it does not require computational skills by the user. The embedded structure comprises a coarse-grained model, enabling the effective exploration of molecular movements and interactions among atoms and molecules in multiprotein systems over time.

My remarks are as follows:

1. In the Introduction section, please include information about existing paid or open-source (desktop) software for simulating disordered proteins or multiprotein systems.

2. The authors' software is described as a black box. In the “2. Materials and Methods” section, please provide details about the structure and functionality of your system. This should include a flowchart of the software architecture and additional development details such as the integrated development environment (IDE), programming languages used, datasets utilized, etc. Step-by-step explanations with screenshots of system usage could also be beneficial.

3. The “4. Discussion” (Conclusions?) section is too brief. Please include a comparison with existing analogs of your system here.

4. A Conclusions section should be added. In this section, along with summarizing the obtained results, please comment on the study limitations and provide directions for future research.

Technical remarks:

“PACSAB” should be defined.

How can users access your web-based system via the https://pacsab.upc.edu web address? Is the described web simulator publicly accessible software?

Author Response

I thank the referee for evaluating the manuscript. I have changed the manuscript in order to satisfy his requirements.

1- I have added the second paragraph in the Introduction section, with information about software for molecular dynamics simulations and web tools available for the study of aggregation of disordered proteins and related topics

2- I have added a flowchart and described thoroughly the functionality of the system. I have specified the programing languages used (PHP, FORTRAN, bash) and the codes used (AmberTools and gnuplot apart form our own simulation and analysis codes implemented in FORTRAN). We do not use any datasets because the simulation method is purely based on a molecular dynamics force field, this has been specified at the beginning of the Methods section.

3- I have added a proper Discussion section, where I compare this web tool with other existing tools, I compare the results obtained here with those of our earlier works, and I comment the limitations of our web tool, that we found for the case of a highly aggregating peptide like AB40.

4- The former "Discussion" section was actually a Conclusions section, since it summarized the results. I have renamed it and I have added the directions for future research.

Answer to the technical remarks:

1- "PACSAB" was defined within the Methods section, I have moved its definition to the Introduction section.

2- Just in case you did not receive from the editors the credentials to access the web tool in the first submission, I send it to you:

User: 'ref2'

Password: 'SCFtcm137'

Round 2

Reviewer 2 Report

Comments and Suggestions for Authors

In this second revision, the authors of ijms-3005135-v2 “PACSAB server: a web-based tool for the study of aggregation and the conformational ensemble of disordered and folded proteins” have addressed all of my concerns.

My recommendation is to accept the manuscript as it is.